# EMERGENT COORDINATION THROUGH COMPETITION

**Siqi Liu,**[*] **Guy Lever,**[*] **Josh Merel, Saran Tunyasuvunakool, Nicolas Heess, Thore Graepel**
DeepMind
London, United Kingdom
{liusiqi,guylever,jsmerel,stunya,heess,thore}@google.com

## ABSTRACT

We study the emergence of cooperative behaviors in reinforcement learning agents by introducing a challenging competitive multi-agent soccer environment with continuous simulated physics. We demonstrate that decentralized, population-based training with co-play can lead to a progression in agents' behaviors: from random, to simple ball chasing, and finally showing evidence of cooperation. Our study highlights several of the challenges encountered in large scale multi-agent training in continuous control. In particular, we demonstrate that the automatic optimization of simple shaping rewards, not themselves conducive to co-operative behavior, can lead to long-horizon team behavior. We further apply an evaluation scheme, grounded by game theoretic principals, that can assess agent performance in the absence of pre-defined evaluation tasks or human baselines.

## 1 INTRODUCTION

Competitive games have been grand challenges for artificial intelligence research since at least the 1950s (Samuel, 1959; Tesauro, 1995; Campbell et al., 2002; Vinyals et al., 2017). In recent years, a number of breakthroughs in AI have been made in these domains by combining deep reinforcement learning (RL) with self-play, achieving superhuman performance at Go and Poker (Silver et al., 2016; Moravčk et al., 2017). In continuous control domains, competitive games possess a natural curriculum property, as observed in Bansal et al. (2017), where complex behaviors have the potential to emerge in simple environments as a result of competition between agents, rather than due to increasing difficulty of manually designed tasks. Challenging collaborative-competitive multi-agent environments have only recently been addressed using end-to-end RL by Jaderberg et al. (2018), which learns visually complex first-person 2v2 video games to human level. One longstanding challenge in AI has been robot soccer (Kitano et al., 1997), including simulated leagues, which has been tackled with machine learning techniques (Riedmiller et al., 2009; MacAlpine & Stone, 2018) but not yet mastered by end-to-end reinforcement learning.

We investigate the emergence of co-operative behaviors through multi-agent competitive games. We design a simple research environment with simulated physics in which complexity arises primarily through competition between teams of learning agents. We introduce a challenging multi-agent soccer environment, using MuJoCo (Todorov et al., 2012) which embeds soccer in a wider universe of possible environments with consistent simulated physics, already used extensively in the machine learning research community (Heess et al., 2016; 2017; Bansal et al., 2017; Brockman et al., 2016; Tassa et al., 2018; Riedmiller et al., 2018). We focus here on multi-agent interaction by using relatively simple bodies with a 3-dimensional action space (though the environment is scalable to more agents and more complex bodies).[1] We use this environment to examine continuous multi-agent reinforcement learning and some of its challenges including coordination, use of shaping rewards, exploitability and evaluation.

We study a framework for continuous multi-agent RL based on decentralized population-based training (PBT) of independent RL learners (Jaderberg et al., 2017; 2018), where individual agents learn off-policy with recurrent memory and decomposed shaping reward channels. In contrast to some recent work where some degree of centralized learning was essential for multi-agent coordinated

---

[*]Equal contribution.
[1]The environment is released at https://git.io/dm_control_soccer.

behaviors (e.g. Lowe et al., 2017; Foerster et al., 2016), we demonstrate that end-to-end PBT can lead to emergent cooperative behaviors in our soccer domain. While designing shaping rewards that induce desired cooperative behavior is difficult, PBT provides a mechanism for automatically evolving simple shaping rewards over time, driven directly by competitive match results. We further suggest to decompose reward into separate weighted channels, with individual discount factors and automatically optimize reward weights and corresponding discounts online. We demonstrate that PBT is able to evolve agents' shaping rewards from myopically optimizing dense individual shaping rewards through to focusing relatively more on long-horizon game rewards, i.e. individual agent's rewards automatically align more with the team objective over time. Their behavior correspondingly evolves from random, through simple ball chasing early in the learning process, to more co-operative and strategic behaviors showing awareness of other agents. These behaviors are demonstrated visually and we provide quantitative evidence for coordination using game statistics, analysis of value functions and a new method of analyzing agents' counterfactual policy divergence.

Finally, evaluation in competitive multi-agent domains remains largely an open question. Traditionally, multi-agent research in competitive domains relies on handcrafted bots or established human baselines (Jaderberg et al., 2018; Silver et al., 2016), but these are often unavailable and difficult to design. In this paper, we highlight that diversity and exploitability of evaluators is an issue, by observing non-transitivities in the agents pairwise rankings using tournaments between trained teams. We apply an evaluation scheme based on *Nash averaging* (Balduzzi et al., 2018) and evaluate our agents based on performance against pre-trained agents in the support set of the *Nash average*.

## 2 PRELIMINARIES

We treat our soccer domain as a *multi-agent reinforcement learning* problem (MARL) which models a collection of agents interacting with an environment and learning, from these interactions, to optimize individual cumulative reward. MARL can be cooperative, competitive or some mixture of the two (as is the case in soccer), depending upon the alignment of agents' rewards. MARL is typically modelled as a *Markov game* (Shapley, 1953; Littman, 1994), which comprises: a *state space* $\mathcal{S}$, $n$ agents with observation and action sets $\mathcal{O}^1, ..., \mathcal{O}^n$ and $\mathcal{A}^1, ..., \mathcal{A}^n$; a (possibly stochastic) reward function $R^i : \mathcal{S} \times \mathcal{A}^i \to \mathbb{R}$ for each agent; observation functions $\phi^i : \mathcal{S} \to \mathcal{O}^i$; a transition function $P$ which defines the conditional distribution over successor states given previous state-actions: $P(S_{t+1}|S_t, A_t^1, ..., A_t^n)$, which satisfies the Markov property $P(S_{t+1}|S_\tau, A_\tau^1, ..., A_\tau^n, \forall \tau \leq t) = P(S_{t+1}|S_t, A_t^1, ..., A_t^n)$; and a start state distribution $P_0(S_0)$ on $\mathcal{S}$. In our application the state and action sets are continuous, and the transition distributions should be thought of as densities. Each agent $i$ sequentially chooses actions, $a_t^i$, at each timestep $t$, based on their observations, $\phi_t^i = \phi^i(s_t)$, and these interactions give rise to a trajectory $((s_t, a_t^1, ..., a_t^n, r_t^1, ..., r_t^n))_{t=1,2,...,H}$, over a horizon $H$, where at each time step $S_{t+1} \sim P(\cdot|s_t, a_t^1, ..., a_t^n)$, and $r_t^i = R^i(s_t, a_t^i)$. Each agent aims to maximize expected cumulative reward, $\mathbb{E}[\sum_{t=0}^{H} \gamma^t r_t^i]$ (discounted by a factor $\gamma < 1$ to ensure convergence when $H$ is infinite), and chooses actions according to a policy $a_t^i \sim \pi^i(\cdot|x_t^i)$, which in general can be any function of the *history* $x_t^i$ of the agent's prior observations and actions at time $t$, $x_t^i := (\phi^i(s_1), a_1^i, ..., \phi^i(s_{t-1}), a_{t-1}^i, \phi^i(s_t))$. The special case of a Markov game with one agent is a partially-observed Markov decision process (POMDP) (Sutton & Barto, 1998). In this work all players have the same action and observation space.

## 3 METHODS

We seek a method of training agents which addresses the exploitability issues of competitive games, arising from overfitting to a single opponents policy, and provides a method of automatically optimizing hyperparameters and shaping rewards online, which are otherwise hard to tune. Following Jaderberg et al. (2018), we combine algorithms for single-agent RL (in our case, *SVG0* for continuous control) with *population-based training* (PBT) (Jaderberg et al., 2017). We describe the individual components of this framework, and several additional novel algorithmic components introduced in this paper.

### 3.1 POPULATION BASED TRAINING

Population Based Training (PBT) (Jaderberg et al., 2017) was proposed as a method to optimize hyperparameters via a population of simultaneously learning agents: during training, poor performing

---

**Algorithm 1** Population-based Training for Multi-Agent RL.

---

1: **procedure** PBT-MARL
2:     $\{A_i\}_{i \in [1,..,N]}$ $N$ independent agents forming a population.
3:     **for** agent $A_i$ in $\{A_i\}_{i \in [1,..,N]}$ **do**
4:         Initialize agent network parameters $\theta_i$ and agent rating $r_i$ to fixed initial rating $R_{init}$.
5:         Sample initial hyper-parameter $\theta_i^h$ from the initial hyper-parameter distribution.
6:     **end for**
7:     **while** true **do**
8:         Agents play *TrainingMatches* and update network parameters by *Retrace-SVG0*.
9:         **for** match result $(s_i, s_j) \in TrainingMatches$ **do**
10:             $UpdateRating(r_i, r_j, s_i, s_j)$                          ▷ See Appendix B.1
11:         **end for**
12:         **for** agent $A_i \in \{A_i\}_{i \in [1,..,N]}$ **do**                 ▷ Evolution Procedure
13:             **if** $Eligible(A_i)$ **then**                              ▷ See Appendix B.2
14:                 $A_j \leftarrow Select(A_i, \{A_i\}_{i \in [1,..,N]; i \neq j})$           ▷ See Appendix B.3
15:                 **if** $A_j \neq$ NULL **then**
16:                     $Inherit(\theta_i, \theta_j, \theta_i^h, \theta_j^h)$          ▷ $A_i$ inherits from $A_j$, See Appendix B.4
17:                     $\theta_i^h \leftarrow Mutate(\theta_i^h)$                      ▷ See Appendix B.5
18:                 **end if**
19:             **end if**
20:         **end for**
21:     **end while**
22: **end procedure**

---

agents, according to some fitness function, inherit network parameters and some hyperparameters from stronger agents, with additional mutation. Hyperparameters can continue to evolve during training, rather than committing to a single fixed value (we show that this is indeed the case in Section 5.1). PBT was extended to incorporate co-play (Jaderberg et al., 2018) as a method of optimizing agents for MARL: subsets of agents are selected from the population to play together in multi-agent games. In any such game each agent in the population effectively treats the other agents as part of their environment and learns a policy $\pi_\theta$ to optimize their expected return, averaged over such games. In any game in which $\pi_\theta$ controls player $i$ in the game, if we denote by $\pi_{\backslash i} := \{\pi^j\}_{j \in \{1,2,...,n\}, j \neq i}$ the policies of the other agents $j \neq i$, we can write the expected cumulative return over a game as

$$J^i(\pi_\theta; \pi_{\backslash i}) := \mathbb{E}\left[\sum_{t=0}^{H} \gamma^t r_t^i \big| \pi^i = \pi_\theta, \pi_{\backslash i}\right] \tag{1}$$

where the expectation is w.r.t. the environment dynamics and conditioned on the actions being drawn from policies $\pi_\theta$ and $\pi_{\backslash i}$. Each agent in the population attempts to optimize (1) averaged over the draw of all agents from the population $\mathcal{P}$, leading to the PBT objective $J(\pi_\theta) := \mathbb{E}_i[\mathbb{E}_{\pi_{\backslash i} \sim \mathcal{P}}[J^i(\pi_\theta; \pi_{\backslash i}) | \pi^i = \pi_\theta]]$, where the outer expectation is w.r.t. the probability that the agent with policy $\pi_\theta$ controls player $i$ in the environment, and the inner expectation is the expectation over the draw of other agents, conditioned on $\pi_\theta$ controlling player $i$ in the game. PBT achieves some robustness to exploitability by training a population of learning agents against each other. Algorithm 1 describes PBT-MARL for a population of $N$ agents $\{A_i\}_{i \in [1,..,N]}$, employed in this work.

## 3.2 RETRACE-SVG0

Throughout our experiments we use Stochastic Value Gradients (SVG0) (Heess et al., 2015b) as our reinforcement learning algorithm for continuous control. This is an actor-critic policy gradient algorithm, which in our setting is used to estimate gradients $\frac{\partial}{\partial \theta} J^i(\pi_\theta; \pi_{\backslash i})$ of the objective (1) for each game. Averaging these gradients over games will effectively optimize the PBT objective $J(\pi_\theta)$. Policies are additionally regularized with an entropy loss $H(\pi)$ i.e. we maximize $\hat{J}(\pi_\theta) := J(\pi_\theta) + \alpha H(\pi_\theta)$ using the Adam optimizer (Kingma & Ba, 2014) to apply gradient updates where $\alpha$ represents a multiplicative entropy cost factor. A derivation of SVG0 is provided in Appendix A.

SVG utilizes a differentiable Q-critic. Our critic is learned using experience replay, minimizing a $k$-step TD-error with off-policy retrace corrections (Munos et al., 2016), using a separate target

network for bootstrapping, as is also described in Hausman et al. (2018); Riedmiller et al. (2018). The identity of other agents $\pi_{\setminus i}$ in a game are not explicitly revealed but are potentially vital for accurate action-value estimation (value will differ when playing against weak rather than strong opponents). Thus, we use a recurrent critic to enable the $Q$-function to implicitly condition on other players observed behavior, better estimate the correct value for the current game, and generalize over the diversity of players in the population of PBT, and, to some extent, the diversity of behaviors in replay. We find in practice that a recurrent $Q$-function, learned from partial unrolls, performs very well. Details of our Q-critic updates, including how memory states are incorporated into replay, are given in Appendix A.2.

### 3.3 Decomposed Discounts and action-value estimation for Reward shaping

Reinforcement learning agents learning in environments with sparse rewards often require additional reward signal to provide more feedback to the optimizer. Reward can be provided to encourage agents to explore novel states for instance (e.g. Brafman & Tennenholtz, 2001), or some other form of intrinsic motivation. Reward shaping is particularly challenging in continuous control (e.g. Popov et al., 2017) where obtaining sparse rewards is often highly unlikely with random exploration, but shaping can perturb objectives (e.g. Bagnell & Ng, 2005) resulting in degenerate behaviors. Reward shaping is yet more complicated in the cooperative multi-agent setting in which independent agents must optimize a joint objective. Team rewards can be difficult to co-optimize due to complex credit assignment, and can result in degenerate behavior where one agent learns a reasonable policy before its teammate, discouraging exploration which could interfere with the first agent's behavior as observed by Hausknecht (2016). On the other hand, it is challenging to design shaping rewards which induce desired co-operative behavior.

We design $n_r$ shaping reward functions $\{r_j : \mathcal{S} \times \mathcal{A} \to \mathbb{R}\}_{j=1,...,n_r}$, weighted so that $r(\cdot) := \sum_{j=1}^{n_r} \alpha_j r_j(\cdot)$ is the agent's internal reward and, as in Jaderberg et al. (2018), we use population-based training to optimize the relative weighting $\{\alpha_j\}_{j=1,...,n_r}$. Our shaping rewards are simple individual rewards to help with exploration, but which would induce degenerate behaviors if badly scaled. Since the fitness function used in PBT will typically be the true environment reward (in our case win/loss signal in soccer), the weighting of shaping rewards can in principle be automatically optimized online using the environment reward signal. One enhancement we introduce is to optimize separate discount factors $\{\gamma_j\}_{j=1,...,n_r}$ for each individual reward channel. The objective optimized is then (recalling Equation 1) $J(\pi_\theta; \pi_{\setminus i}) := \mathbb{E}\big[\sum_{j=1}^{n_r} \alpha_j \sum_{t=0}^{H} \gamma_j^t r_j(s_t, a_t^1, ..., a_t^n)\big|\pi^i = \pi_\theta, \pi_{\setminus i}\big]$. This separation of discount factors enables agents to learn to optimize the sparse environment reward far in the future with a high discount factor, but optimize dense shaping rewards myopically, which would also make value-learning easier. This would be impossible if discounts were confounded. The specific shaping rewards used for soccer are detailed in Section 5.1.

## 4 Experimental Setup

### 4.1 MuJoCo Soccer Environment

We simulate 2v2 soccer using the MuJoCo physics engine (Todorov et al., 2012). The 4 players in the game are a single sphere (the body) with 2 fixed arms, and a box head, and have a 3-dimensional action space: accelerate the body forwards/backwards, torque can be applied around the vertical axis to rotate, and apply downwards force to "jump". Applying torque makes the player spin, gently for steering, or with more force in order to "kick" the football with its arms. At each timestep, proprioception (position, velocity, accelerometer information), task (egocentric ball position, velocity and angular velocity, goal and corner positions) and teammate and opponent (orientation, position and velocity) features are observed making a 93-dimensional input observation vector. Each soccer match lasts upto 45 seconds, and is terminated when the first team scores. We disable contacts between the players, but enable contacts between the players, the pitch and the ball. This makes it impossible for players to foul and avoids the need for a complicated contact rules, and led to more dynamic matches. There is a small border around the pitch which players can enter, but when the ball is kicked out-of-bounds it is reset by automatic "throw in" a small random distance towards the center of the pitch, and no penalty is incurred. The players choose a new action every 0.05 seconds. At the start of an episode the players and ball are positioned uniformly at random on the pitch. We train agents on a field whose dimensions are randomized in the range $20m \times 15m$ to $28m \times 21m$,

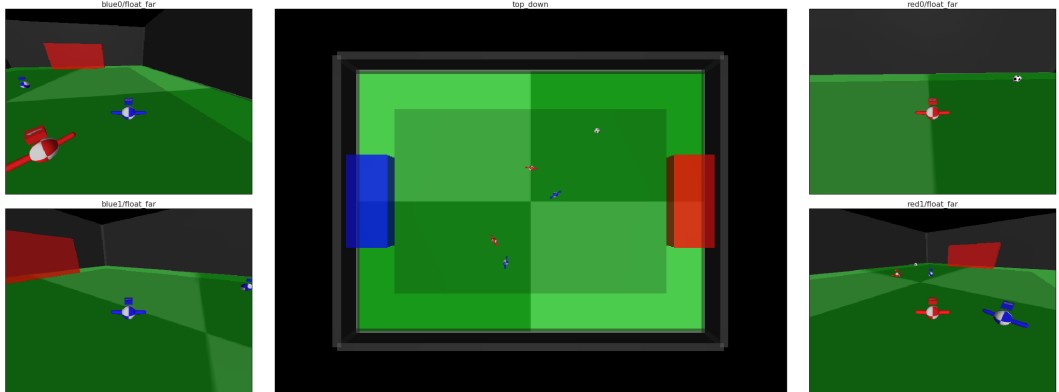

Figure 1: Top-down view with individual camera views of 2v2 multi-agent soccer environment.

with fixed aspect ratio, and are tested on a field of fixed size $24m \times 18m$. We show an example frame of the game in Figure 1.

## 4.2 PBT SETTINGS

We use population-based training with 32 agents in the population, an agent is chosen for evolution if its expected win rate against another chosen agent drops below 0.47. The k-factor learning rate for Elo is 0.1 (this is low, due to the high stochasticity in the game results). Following evolution there is a grace period where the agent does not learn while its replay buffer refills with fresh data, and a further "burn-in" period before the agent can evolve again or before its weights can be copied into another agent, in order to limit the frequency of evolution and maintain diversity in the population. For each 2v2 training match 4 agents were selected uniformly at random from the population of 32 agents, so that agents are paired with diverse teammates and opponents.

## 4.3 EVALUATION

Unlike multi-agent domains where we possess hand-crafted bots or human baselines, evaluating agent performance in novel domains where we do not possess such knowledge remains an open question. A number of solutions have been proposed: for competitive board games, there exits evaluation metrics such as Elo (Elo, 1978) where ratings of two players should translate to their relative win-rates; in professional team sports, head-to-head tournaments are typically used to measure team performance; in Al-Shedivat et al. (2017), survival-of-the-fittest is directly translated to multi-agent learning as a proxy to relative agent performance. Unfortunately, as shown in Balduzzi et al. (2018), in a simple game of rock-paper-scissors, a rock-playing agent will attain high Elo score if we simply introduce more scissor-play agents into a tournament. Survival-of-the-fittest analysis as shown in Al-Shedivat et al. (2017) would lead to a cycle, and agent ranking would depend on when measurements are taken (Tuyls et al., 2018).

**Nash-Averaging Evaluators:** One desirable property for multi-agent evaluation is **invariance to redundant agents**: i.e. the presence of multiple agents with similar strategies should not bias the ranking. In this work, we apply Nash-averaging which possesses this property. Nash-Averaging consists of a meta-game played using a pair-wise win-rate matrix between N agents. A *row* player and a *column* player simultaneously pick distributions over agents for a mixed strategy, aiming for a non-exploitable strategy (see Balduzzi et al., 2018).

In order to meaningfully evaluate our learned agents, we need to bootstrap our evaluation process. Concretely, we choose a set of fixed evaluation teams by Nash-averaging from a population of 10 teams previously produced by diverse training schemes, with 25B frames of learning experience each. We collected 1M tournament matches between the set of 10 agents. Figure 2 shows the pair-wise expected goal difference among the 3 agents in the support set. Nash Averaging assigned non-zero weights to 3 teams that exhibit diverse policies with non-transitive performance which would not have been apparent under alternative evaluation schemes: agent A wins or draws against agent B on 59.7% of the games; agent B wins or draws against agent C on 71.1% of the games and agent C

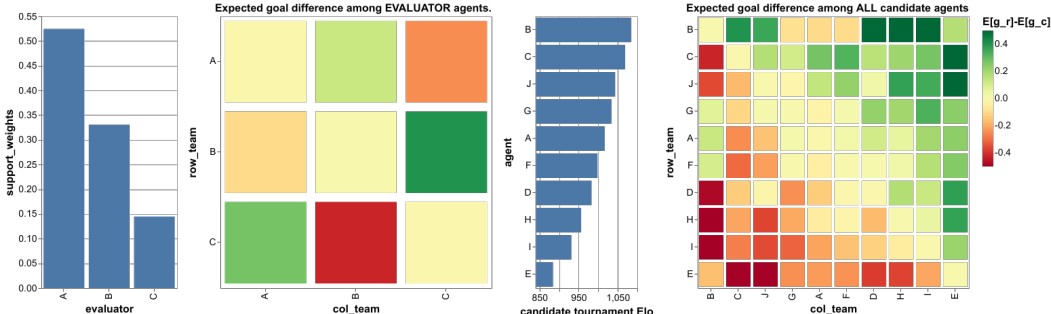

Figure 2: *L1:* selected set of agents in Nash support set with their respective support weights. *L2:* pair-wise expected goal difference among **evaluator** agents. *L3:* Elo ratings for **all** agents computed from tournament matches. *L4:* pair-wise expected goal difference among **all** agents.

wins or draws against agent A on 65.3% of the matches. We show recordings of example tournament matches between agent A, B and C to demonstrate qualitatively the diversity in their policies (video 3 on the website [2]). Elo rating alone would yield a different picture: agent *B* is the best agent in the tournament with an Elo rating of 1084.27, followed by *C* at 1068.85; Agent *A* ranks 5th at 1016.48 and **we would have incorrectly concluded that agent *B* ought to beat agent *A* with a win-rate of 62%.** All variants of agents presented in the experimental section are evaluated against the set of 3 agents in terms of their pair-wise expected difference in score, weighted by support weights.

## 5 RESULTS

We describe in this section a set of experimental results. We first present the incremental effect of various algorithmic components. We further show that population-based training with co-play and reward shaping induces a progression from *random* to *simple ball chasing* and finally *coordinated* behaviors. A tournament between all trained agents is provided in Appendix D.

### 5.1 ABLATION STUDY

We incrementally introduce algorithmic components and show the effect of each by evaluating them against the set of 3 evaluation agents. We compare agent performance using expected goal difference weighted according to the Nash averaging procedure. We annotate a number of algorithmic components as follows: **ff**: feedforward policy and action-value estimator; **evo**: population-based training with agents evolving within the population; **rwd_shp**: providing dense shaping rewards on top of sparse environment scoring/conceding rewards; **lstm**: recurrent policy with recurrent action-value estimator; **lstm_q**: feedforward policy with recurrent action-value estimator; **channels**: decomposed action-value estimation for each reward component; each with its own, individually evolving discount factor.

**Population-based Training with Evolution:** We first introduce PBT with evolution. Figure 3 (**ff** vs **ff + evo**) shows that Evolution kicks in at 2B steps, which quickly improves agent performance at the population level. We show in Figure 4 that Population-based training coupled with evolution yields a natural progression of learning rates, entropy costs as well as the discount factor. Critic learning rate gradually decreases as training progresses, while discount factor increases over time, focusing increasingly on long-term return. Entropy costs slowly decreases which reflects a shift from exploration to exploitation over the course training.

**Reward Shaping:** We introduced two simple dense shaping rewards in addition to the sparse scoring and conceding environment rewards: *vel-to-ball*: player's linear velocity projected onto its unit direction vector towards the ball, thresholded at zero; *vel-ball-to-goal*: ball's linear velocity projected onto its unit direction vector towards the center of opponent's goal. Furthermore the sparse *goal* reward and *concede* penalty are separately evolved, and so can receive separate weight that trades off between the importance of scoring versus conceding.

---

[2] https://goo.gl/AuHv7V

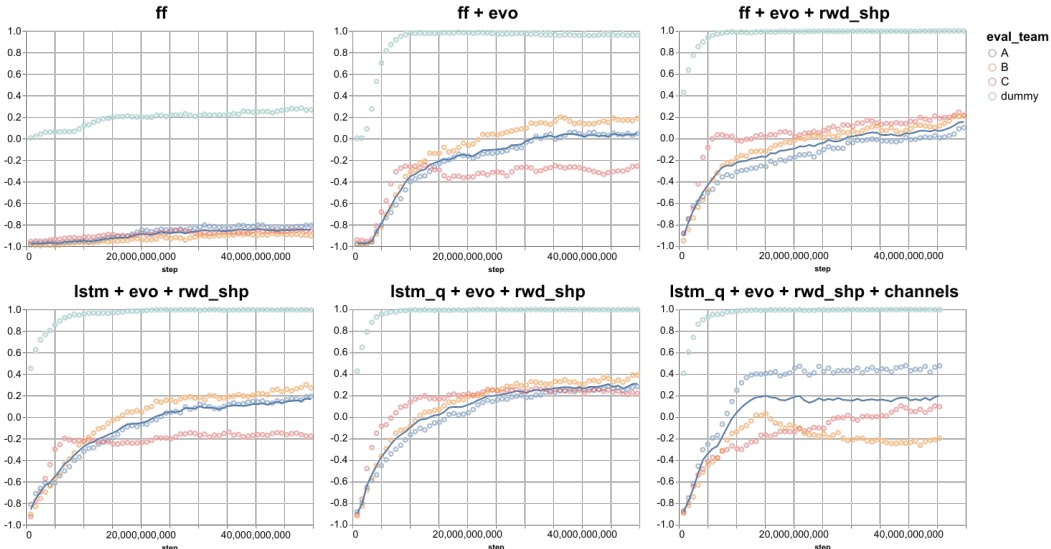

Figure 3: Weighted expected goal difference shown in blue line. Agents' expected goal difference against each evaluator agent in point plot. A dummy evaluator that takes random actions has been introduced to show learning progress early in the training, with zero weight in the performance computation.

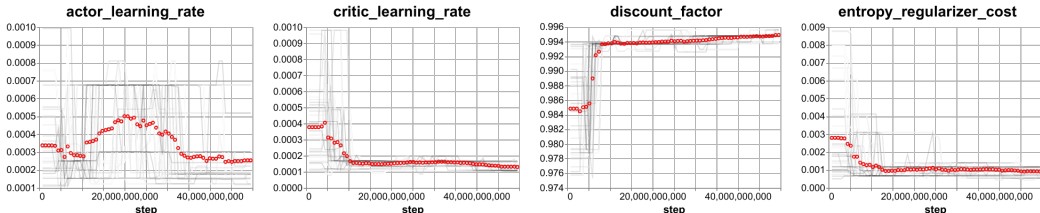

Figure 4: Evolution of hyper-parameters. Hyperparameters of individual agents within the population in gray.

Dense shaping rewards make learning significantly easier early in training. This is reflected by agents' performance against the dummy evaluator where agents with dense shaping rewards quickly start to win games from the start (Figure 3, **ff + evo** vs **ff + evo + rwd_shp**). On the other hand, shaping rewards tend to induce sub-optimal policies (Ng et al., 1999; Popov et al., 2017); We show in Figure 5 however that this is mitigated by coupling training with hyper-parameter evolution which adaptively adjusts the importance of shaping rewards. Early on in the training, the population as a whole decreases the penalty of conceding a goal which evolves towards zero, assigning this reward relatively lower weight than scoring. This trend is subsequently reversed towards the end of training, where the agents evolved to pay more attention to conceding goals: i.e. agents first learn to optimize scoring and then incorporate defending. The dense shaping reward *vel-to-ball* however quickly decreases in relative importance which is mirrored in their changing behavior, see Section 5.2.

**Recurrence:** The introduction of recurrence in the action-value function has a significant impact on agents' performance as shown in Figure 3 (**ff + evo + rwd_shp** vs **lstm + evo + rwd_shp** reaching weighted expected goal difference of 0 at 22B vs 35B steps). A recurrent policy seems to underperform its feedforward counterpart in the presence of a recurrent action-value function. This could be due to out-of-sample evaluators which suggests that recurrent policy might overfit to the behaviors of agents from its own population while feedforward policy cannot.

**Decomposed Action-Value Function:** While we observed empirically that the discount factor increases over time during the evolution process, we hypothesize that different reward components require different discount factor. We show in Figure 6 that this is indeed the case, for sparse environment rewards and *vel-ball-to-goal*, the agents focus on increasingly long planning horizon. In contrast, agents quickly evolve to pay attention to short-term returns on *vel-to-ball*, once they learned the basic movements. Note that although this agent underperforms **lstm + evo + rwd_shp** asymptot-

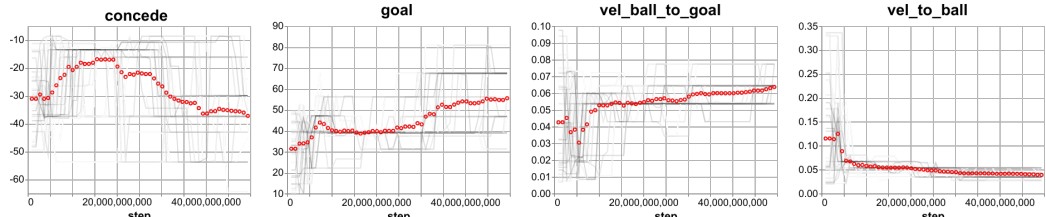

Figure 5: Evolution of relative importance of dense shaping rewards over the course of training. Hyperparameters of individual agents within the population in gray.

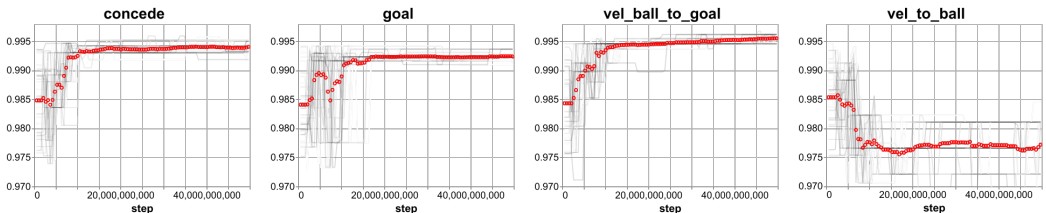

Figure 6: Evolution of discount factor for each reward component. We show hyperparameters of individual agents within the population in gray.

ically, it achieved faster learning in comparison (reaching 0.2 at 15B vs 35B). This agent also attains the highest Elo in a tournament between all of our trained agents, see Appendix D. This indicates that the training population is less diverse than the Nash-averaging evaluation set, motivating future work on introducing diversity as part of training regime.

## 5.2 EMERGENT MULTI-AGENT BEHAVIORS

Assessing cooperative behavior in soccer is difficult. We present several indicators ranging from behavior statistics, policy analysis to behavior probing and qualitative game play in order to demonstrate the level of cooperation between agents.

We provide birds-eye view videos on the website[2] (video 1), where each agent's value-function is also plotted, along with a bar plot showing the value-functions for each weighted shaping reward component. Early in the matches the 2 dense shaping rewards (rightmost channels) dominate the value, until it becomes apparent that one team has an advantage at which point all agent's value functions become dominated by the sparse conceding/scoring reward (first and second channels) indicating that PBT has learned a balance between sparse environment and dense shaping rewards so that positions with a clear advantage to score will be preferred. There are recurring motifs in the videos: for example, evidence that agents have learned a "cross" pass from the sideline to a teammate in the centre (see Appendix F for example traces), and frequently appear to anticipate this and change direction to receive. Another camera angle is provided on the website[2] (video 2) showing representative, consecutive games played between two fixed teams. These particular agents generally kick the ball upfield, avoiding opponents and towards teammates.

### 5.2.1 BEHAVIOR STATISTICS

Statistics collected during matches are shown in Figure 7. The **vel-to-ball** plot shows the agents average velocity towards the ball as training progresses: early in the learning process agents quickly maximize their velocity towards the ball (optimizing their shaping reward) but gradually fixate less on simple ball chasing as they learn more useful behaviors, such as kicking the ball upfield. The **teammate-spread-out** shows the evolution of the spread of teammates position on the pitch. This shows the percentage of timesteps where the teammates are spread at least 5m apart: both agents quickly learn to hog the ball, driving this lower, but over time learn more useful behaviors which result in diverse player distributions. **pass/interception** shows that *pass*, where players from the same team consecutively kicked the ball and *interception*, where players from the opposing teams kicked the ball in sequence, both remain flat throughout training. To *pass* is the more difficult

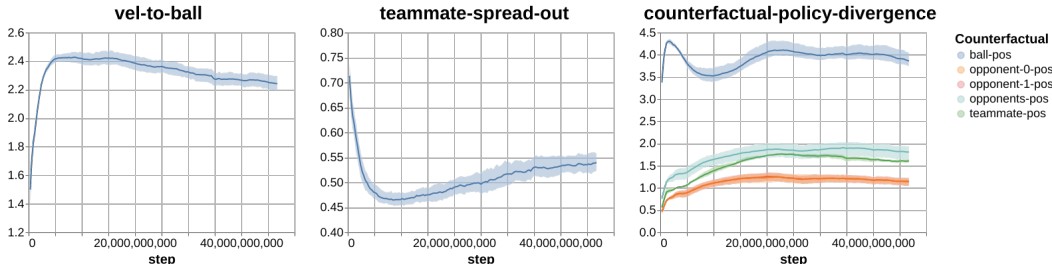

Figure 7: Behavior statistics evolution.

Figure 8: *L1:* agent's average velocity towards the ball. *L2:* percentage of time when players within a team are spread out. *L3:* KL divergence incurred by replacing a subset of state with counterfactual information.

behavior as it requires two teammates to coordinate whereas interception only requires one of the two opponents to position correctly. **pass/interception-10m** logs pass/interception events over more than 10m, and here we see a dramatic increase in *pass-10m* while *interception-10m* remains flat, i.e. long range passes become increasingly common over the course of training, reaching equal frequency as long-range interception.

### 5.2.2 COUNTERFACTUAL POLICY DIVERGENCE

In addition to analyzing behavior statistics, we could ask the following: *"had a subset of the observation been different, how much would I have changed my policy?"*. This reveals the extent to which an agent's policy is dependent on this subset of the observation space. To quantify this, we analyze counterfactual policy divergence: at each step, we replace a subset of the observation with 10 valid alternatives, drawn from a fixed distribution, and we measure the KL divergence incurred in agents' policy distributions. This cannot be measured for a recurrent policy due to recurrent states and we investigate **ff + evo + rwd_shp** instead (Figure 3), where the policy network is feedforward. We study the effect of five types of counterfactual information over the course of training.

*ball-position* has a strong impact on agent's policy distribution, more so than player and opponent positions. Interestingly, *ball-position* initially reaches its peak quickly while divergence incurred by counterfactual player/opponent positions plateau until reaching 5B training steps. This phase coincides with agent's greedy optimization of shaping rewards, as reflected in Figure 8. Counterfactual *teammate/opponent position* increasingly affect agents' policies from 5B steps, as they spread out more and run less directly towards the ball. *Opponent-0/1-position* incur less divergence than teammate position individually, suggesting that teammate position has relatively large impact than any single opponent, and increasingly so during 5B-20B steps. This suggests that comparatively players learn to leverage a coordinating teammate first, before paying attention to competing opponents. The gap between *teammate-position* and *opponents-position* eventually widens, as opponents become increasingly relevant to the game dynamics. The progression observed in counterfactual policy divergence provides evidence for emergent cooperative behaviors among the players.

### 5.2.3 MULTI-AGENT BEHAVIOR PROBING

Qualitatively, we could ask the following question: *would agents coordinate in scenarios where it's clearly advantageous to do so?* To this end, we designed a probe task, to test our trained agents for coordination, where *blue0* possesses the ball, while the two opponents are centered on the pitch in front. A teammate *blue1* is introduced to either left or right side. In Figure 9 we show typical traces of agents' behaviors (additional probe task video shown at Video 4 on our website[2]): at 5B steps,

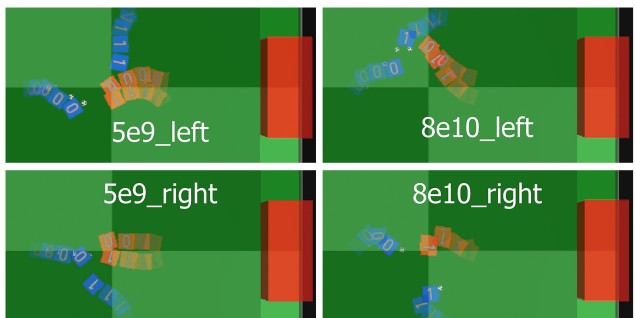

| | pass | intercept |
|---|---|---|
| 5B_left | 0 | 100 |
| 5B_right | 31 | 90 |
| 80B_left | 76 | 24 |
| 80B_right | 56 | 27 |

Figure 9: *L1:* Comparison between two snapshots (5B vs 80B) of the same agent. *L2:* number of successful passes and interception occurred in the first 100 timesteps, aggregated over 100 episodes.

when agents play more individualistically, we observe that *blue0* always tries to dribble the ball by itself, regardless of the position of *blue1*. Later on in the training, *blue0* actively seeks to pass and its behavior is driven by the configuration of its teammate, showing a high-level of coordination. In "8e10_left" in particular, we observe two consecutive pass (*blue0* to *blue1* and back), in the spirit of 2-on-1 passes that emerge frequently in human soccer games.

## 6    RELATED WORK

The population-based training we use here was introduced by Jaderberg et al. (2018) for the capture-the-flag domain, whereas our implementation is for continuous control in simulated physics which is less visually rich but arguably more open-ended, with potential for sophisticated behaviors generally and allows us to focus on complex multi-agent interactions, which may often be physically observable and interpretable (as is the case with passing in soccer). Other recent related approaches to multi-agent training include PSRO (Lanctot et al., 2017) and NFSP (Heinrich & Silver, 2016), which are motivated by game-theoretic methods (fictitious play and double oracle) for solving matrix games, aiming for some robustness by playing previous best response policies, rather than the (more data efficient and parallelizable) approach of playing against simultaneous learning agents in a population. The RoboCup competition is a grand challenge in AI and some top-performing teams have used elements of reinforcement learning (Riedmiller et al., 2009; MacAlpine & Stone, 2018), but are not end-to-end RL. Our environment is intended as a research platform, and easily extendable along several lines of complexity: complex bodies; more agents; multi-task, transfer and continual learning. Coordination and cooperation has been studied recently in deepRL in, for example, Lowe et al. (2017); Foerster et al. (2018; 2016); Sukhbaatar et al. (2016); Mordatch & Abbeel (2018), but all of these require some degree of centralization. Agents in our framework perform fully independent asynchronous learning yet demonstrate evidence of complex coordinated behaviors. Bansal et al. (2017); Al-Shedivat et al. (2017) introduce a MuJoCo Sumo domain with similar motivation to ours, and observe emergent complexity from competition, in a 1v1 domain. We are explicitly interested in cooperation within teams as well as competition. Other attempts at optimizing rewards for multi-agent teams include Liu et al. (2012).

## 7    CONCLUSIONS AND FUTURE WORK

We have introduced a new 2v2 soccer domain with simulated physics for continuous multi-agent reinforcement learning research, and used competition between agents in this simple domain to train teams of independent RL agents, demonstrating coordinated behavior, including repeated passing motifs. We demonstrated that a framework of distributed population-based-training with continuous control, combined with automatic optimization of shaping reward channels, can learn in this environment end-to-end. We introduced the idea of automatically optimizing separate discount factors for the shaping rewards, to facilitate the transition from myopically optimizing shaping rewards towards alignment with the sparse long-horizon team rewards and corresponding cooperative behavior. We have introduced novel method of counterfactual policy divergence to analyze agent behavior. Our evaluation has highlighted non-transitivities in pairwise match results and the practical need for robustness, which is a topic for future work. Our environment can serve as a platform for multi-agent research with continuous physical worlds, and can be easily scaled to more agents and more complex bodies, which we leave for future research.

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

## A  OFF-POLICY SVG0 ALGORITHM

### A.1  POLICY UPDATES

The Stochastic Value Gradients (SVG0) algorithm used throughout this work is a special case of the family of policy gradient algorithms provided by Heess et al. (2015b) in which the gradient of a value function used to compute the policy gradient, and is closely related to the Deterministic Policy Gradient algorithm (DPG) (Silver et al., 2014), which is itself a special case of SVG0. For clarity we provide the specific derivation of SVG0 here.

Using the reparametrization method of Heess et al. (2015b) we write a stochastic policy $\pi_\theta(\cdot|s)$ as a deterministic policy $\mu_\theta : \mathcal{S} \times \mathbb{R} \to \mathcal{A}$ further conditioned on a random variable $\eta \in \mathbb{R}^p$, so that $a \sim \pi_\theta(\cdot|s)$ is equivalent to $a \sim \mu_\theta(s, \eta)$, where $\eta \sim \rho$ for some distribution $\rho$. Then,

$$
\begin{aligned}
Q^{\pi_\theta}(s, a) &= r(s, a) + \gamma \mathbb{E}_{s' \sim P(\cdot|s,a)} \left[ \mathbb{E}_{a' \sim \pi(\cdot|s')} \left[ Q^{\pi_\theta}(s', a') \right] \right] \\
&= r(s, a) + \gamma \mathbb{E}_{s' \sim P(\cdot|s,a)} \left[ \mathbb{E}_{\eta' \sim \rho} \left[ Q^{\pi_\theta}(s', \mu_\theta(s', \eta')) \right] \right]
\end{aligned}
$$

$$
\frac{\partial Q^{\pi_\theta}(s, a)}{\partial \theta} = \gamma \mathbb{E}_{s' \sim P(\cdot|s,a)} \left[ \mathbb{E}_{\eta' \sim \rho} \left[ \frac{\partial}{\partial \theta} Q^{\pi_\theta}(s', \mu_\theta(s', \eta')) \right] \right]
$$

$$
= \gamma \mathbb{E}_{s' \sim P(\cdot|s,a)} \left[ \mathbb{E}_{\eta' \sim \rho} \left[ \frac{\partial}{\partial \theta} Q^{\pi_\theta}(s', a') \Big|_{a'=\mu_\theta(s',\eta')} + \frac{\partial}{\partial a'} Q^{\pi_\theta}(s', a') \Big|_{a'=\mu_\theta(s',\eta')} \frac{\partial}{\partial \theta} \mu_\theta(s', \eta') \right] \right]
$$

from which we obtain a recursion for $\frac{\partial Q^{\pi_\theta}(s,a)}{\partial \theta}$. Expanding the recursion we obtain the policy gradient

$$
\begin{aligned}
\frac{\partial Q^{\pi_\theta}(s_0, a_0)}{\partial \theta} &= \sum_{t=1}^{\infty} \gamma^t \mathbb{E}_{s_t \sim P(\cdot|s_{t-1}, a_{t-1})} \left[ \mathbb{E}_{\eta_t \sim \rho} \left[ \frac{\partial}{\partial a_t} Q^{\pi_\theta}(s_t, a_t) \Big|_{a_t=\mu_\theta(s_t, \eta_t)} \times \right. \right. \\
&\qquad \left. \left. \frac{\partial}{\partial \theta} \mu_\theta(s_t, \eta_t) \right] \Big| s_0, a_0, a_\tau = \mu_\theta(s_\tau, \eta_\tau), \eta_\tau \sim \rho \forall \tau < t \right] \\
&= \int_{\mathcal{S}} \int_{\mathbb{R}^p} \zeta(s, \eta) \frac{\partial}{\partial a} Q^{\pi_\theta}(s, a) \Big|_{a=\mu_\theta(s, \eta)} \frac{\partial}{\partial \theta} \mu_\theta(s, \eta) d\eta ds
\end{aligned}
$$

where $\zeta(s, \eta) := \sum_{t=1}^{\infty} \gamma^t p_t(s, \eta)$ and where $p_t(s, \eta)$ is the joint density over (state, $\eta$) at timestep $t$ following the policy. Typically $\gamma$ is replaced with 1 in the definition of $\zeta$ to avoid discounting terms depending on future states in the gradient too severely. This suggests Algorithm 3 given in Heess et al. (2015b). For details on recurrent policies see Heess et al. (2015a).

---

**Algorithm 2** Off-policy SVG0 algorithm (Heess et al., 2015b).

---

1:  initialize replay buffer $\mathcal{B} = \emptyset$
2:  sample initial state $s_0$ from environment
3:  **for** t **do**=0 to $\infty$ do
4:      sample action from current policy $a_t = \mu_\theta(\cdot|s_t, \eta), \eta \sim \rho$
5:      observe reward and state observation $r_t, s_t$
6:      train $Q^{\pi_\theta}(\cdot, \cdot; \psi)$ off policy using $\mathcal{B}$ (see Section A.2)          ▷ Critic update.
7:      $\theta \leftarrow \theta + \alpha \frac{\partial}{\partial a} Q^{\pi_\theta}(s_t, a; \psi)\big|_{a=\mu_\theta(s_t, \eta_t)} \frac{\partial}{\partial \theta} \mu_\theta(s_t, \eta_t)$          ▷ Policy update.
8:  **end for**

---

### A.2  Q-VALUE UPDATES

As in Section 3.1, in any given game, by treating all other players as part of the environment dynamics, we can define action-value function for policy $\pi_\theta$ controlling player $i$:

$$
Q^{\pi_\theta, i}(s, a; \pi_{\setminus i}) := \mathbb{E}\Big[ \sum_{t=0}^{H} \gamma^t r_t^i \Big| s_0 = s, a_0^i = a; \pi^i = \pi_\theta, \pi_{\setminus i} \Big]
$$

In our soccer environment the reward is invariant over player and we can drop the dependence on $i$.

SVG requires the critic to learn a differentiable Q-function. The true state of the game $s$ and the identity of other agents $\pi_{\backslash i}$, are not revealed during a game and so identities must be inferred from their behavior, for example. Further, as noted in Foerster et al. (2017), off-policy replay is not always fully sound in multi-agent environments since the effective dynamics from any single agent's perspective changes as the other agent's policies change. Because of this, we generally model $Q$ as a function of an agents history of observations - typically keeping a low dimensional summary in the internal state of an LSTM: $Q^{\pi_\theta}(\cdot, \cdot; \psi) : \mathcal{X} \times \mathcal{A} \to \mathbb{R}$, where $\mathcal{X}$ denotes the space of possible histories or internal memory state, parameterized by a neural network with weights $\psi$. This enables the $Q$-function to implicitly condition on other players observed behavior and generalize over the diversity of players in the population and diversity of behaviors in replay, $Q$ is learned using trajectory data stored in an experience replay buffer $\mathcal{B}$, by minimizing the $k$-step return TD-error with off-policy retrace correction (Munos et al., 2016), using a separate target network for bootstrapping, as is also described in Hausman et al. (2018); Riedmiller et al. (2018). Specifically we minimize:

$$L(\psi) := \mathbb{E}_{\xi \sim \mathcal{B}} \left[ (Q^{\pi_\theta}(x_i, a_i; \psi) - Q_{\texttt{retrace}}(\xi))^2 \right]$$

where $\xi := ((s_t, a_t, r_t))_{t=i}^{i+k}$ is a k-step trajectory snippet, where $i$ denotes the timestep of the first state in the snippet, sampled uniformly from the replay buffer $\mathcal{B}$ of prior experience, and $Q_{\texttt{retrace}}$ is the off-policy corrected retrace target:

$$Q_{\texttt{retrace}}(\xi) := \hat{Q}(x_i, a_i; \hat{\psi}) + \sum_{t=0}^{k} \gamma^t \left( \prod_{s=i+1}^{t+i} c_s \right) \left( r(s_{i+t}, a_{i+t}) + \right.$$

$$\left. \gamma \mathbb{E}_{a \sim \hat{\pi}(\cdot|x_{i+t+1})} [\hat{Q}(x_{i+t+1}, a; \hat{\psi})] - \hat{Q}(x_{i+t}, a_{i+t}; \hat{\psi}) \right)$$

where, for stability, $\hat{Q}(\cdot, \cdot; \hat{\psi}) : \mathcal{X} \times \mathcal{A} \to \mathbb{R}$ and $\hat{\pi}$ are *target* network and policies (Mnih et al., 2015) periodically synced with the online action-value critic and policy (in our experiments we sync after every 100 gradient steps), and $c_s := min(1, \frac{\pi(a_s|x_s)}{\beta(a_s|x_s)})$, where $\beta$ denotes the *behavior policy* which generated the trajectory snippet $\xi$ sampled from $\mathcal{B}$, and $\prod_{s=i+1}^{i} c_s := 1$. In our soccer experiments $k = 40$. Though we use off-policy corrections, the replay buffer has a threshold, to ensure that data is relatively recent.

When modelling $Q$ using an LSTM the agent's internal memory state at the first timestep of the snippet is stored in replay, along with the trajectory data. When replaying the experience the LSTM is primed with this stored internal state but then updates its own state during replay of the snippet. LSTMs are optimized using backpropagation through time with unrolls truncated to length 40 in our experiments.

## B  POPULATION-BASED TRAINING PROCEDURE

### B.1  FITNESS

We use Elo rating (Elo (1978)), introduced to evaluate the strength of human chess players, to measure an agent's performance within the population of learning agents and determine eligibility for evolution. Elo is updated from pairwise match results and can be used to predict expected win rates against the other members of the population.

For a given pair of agents $i, j$ (or a pair of agent teams), $s_{elo}$ estimates the expected win rate of agent $i$ playing against agent $j$. We show in Algorithm 3 the update rule for a two player competitive game for simplicity, for a team of multiple players, we use their average Elo score instead.

By using Elo as the fitness function, driving the evolution of the population's hyperparamters, the agents' internal hyperparameters (see Section 3.3) can be automatically optimized for the objective we are ultimately interested in - the win rate against other agents. Individual shaping rewards would otherwise be difficult to handcraft without biasing this objective.

---

**Algorithm 3** Iterative Elo rating update.

1: Initialize rating $r_i$ for each agent in the agent population.
2: $K$: step size of Elo rating update given one match result.
3: $s_i, s_j$: score for agent $i, j$ in a given match.
4: **procedure** UPDATERATING($r_i, r_j, s_i, s_j$)
5:     $s \leftarrow (\text{sign}(s_i - s_j) + 1)/2$
6:     $s_{elo} \leftarrow 1/(1 + 10^{(r_j - r_i)/400})$
7:     $r_i \leftarrow r_i + K(s - s_{elo})$
8:     $r_j \leftarrow r_j - K(s - s_{elo})$
9: **end procedure**

---

## B.2 EVOLUTION ELIGIBILITY

To limit the frequency of evolution and prevent premature convergence of the population, we adopted the same eligibility criteria introduced in Jaderberg et al. (2017). In particular, we consider an agent $i$ eligible for evolution if it has:

1. processed $2 \times 10^9$ frames for learning since the beginning of training; and
2. processed $4 \times 10^8$ frames for learning since the last time it became eligible for evolution.

and agent $j$ can be a parent if agent $j$ has

1. processed $4 \times 10^8$ frames for learning since it last evolved.

which we refer to as a "burn-in" period.

## B.3 SELECTION

When an agent $i$ becomes eligible for evolution, it is compared against another agent $j$ who has finished its "burn-in" period for evolution selection. We describe this procedure in Algorithm 4.

---

**Algorithm 4** Given agent $i$, select an agent $j$ to evolve to.

1: $T_{select}$: win rate selection threshold below which $A_i$ should evolve to $A_j$.
2: $r_i, r_j$: Elo ratings of agents $i, j$.
3: **procedure** SELECT($A_i, \{A_i\}_{i \in [1,..,N]; i \neq j}$)
4:     Choose $A_j$ uniformly at random from $\{A_i\}_{i \in [1,..,N]; i \neq j}$.
5:     $s_{elo} \leftarrow 1/(1 + 10^{(r_j - r_i)/400})$
6:     **if** $s_{elo} < T_{select}$ **then**
7:         **return** $A_j$
8:     **else**
9:         **return** NULL
10:     **end if**
11: **end procedure**

---

## B.4 INHERITANCE

Upon selection for evolution, agent $i$ *inherits* hyperparameters from agent $j$ by "cross-over" meaning that hyperparameters are either inherited or not independently with probability 0.5 as described in Algorithm 5:

## B.5 MUTATION

Upon each evolution action the child agent mutates its hyper-parameters with mutation probability $p_{mutate}$ at a multiplicative perturbation scale $p_{perturb}$. In this work, we apply a mutation probability of $p_{mutate} = 0.1$ and $p_{perturb} = 0.2$ for all experiments. We limit a subset of hyperparameters to bounded ranges (e.g. discount factor) such that their values remain valid throughout training.

---

**Algorithm 5** Agent $i$ inherits from agent $j$ by cross-over.

1: Agent $i, j$ with respective network parameters $\theta_i, \theta_j$ and hyper-parameters $\theta_i^h, \theta_j^h$.
2: **procedure** INHERIT($\theta_i, \theta_j, \theta_i^h, \theta_j^h$)
3: $\quad \theta_i \leftarrow \theta_j$
4: $\quad \boldsymbol{m} = (m_k)_k, \quad m_k \sim \texttt{bernouilli}(0.5)$
5: $\quad \theta_i^h \leftarrow \boldsymbol{m}\theta_i^h + (1 - \boldsymbol{m})\theta_j^h$
6: **end procedure**

---

## C FURTHER ENVIRONMENT DETAILS AND AGENT PARAMETERIZATION

### C.1 POLICY PARAMETRIZATION AND OPTIMIZATION

We parametrize each agent's policy and critic using neural networks. Observation preprocessing is first applied to each raw teammate and opponent feature using a shared 2-layer network with 32 and 16 neurons and Elu activations (Clevert et al., 2015) to embed each individual player's data into a consistent, learned 16 dimensional embedding space. The maximum, minimum and mean of each dimension is then passed as input to the remainder of the network, where it is concatenated with the ball and pitch features. This preprocessing makes the network architecture invariant to the order of teammates and opponents features.

Both critic and actor then apply 2 feed-forward, elu-activated, layers of size 512 and 256, followed by a final layer of 256 neurons which is either feed-forward or made recurrent using an LSTM (Hochreiter & Schmidhuber, 1997). Weights are not shared between critic and actor networks.

We learn the parametrized gaussian policies using SVG0 as detailed in Appendix A, and the critic as described in Section A.2, with the Adam optimizer (Kingma & Ba, 2014) used to apply gradient updates.

## D HEAD-TO-HEAD TOURNAMENT OF TRAINED AGENTS

We also ran a round robin tournament with 50,000 matches between the best teams from 5 populations of agents (selected by Elo within their population), all trained for 5e10 agent steps - i.e. each learner had processed at least 5e10 frames from the replay buffer, though the number of raw environment steps would be much lower than that) and computed the Elo score. This shows the advantage of including shaping rewards, adding a recurrent critic and separate reward and discount channels, and the further (marginal) contribution of a recurrent actor. The full win rate matrix for this tournament is given in Figure 10. Note that the agent with full recurrence and separate reward channels attains the highest Elo in this tournament, though performance against our Nash evaluators in Section 5.1 is more mixed. This highlights the possibility for non-transitivities in this domain and the practical need for robustness to opponents.

## E HYPERPARAMETER EVOLUTION

To assess the relative importance of hyperparameters we replicated a single experiment (using a feed-forward policy and critic network) with 3 different seeds, see Figure 11. Critic learning rate and entropy regularizer evolve consistently over the three training runs. In particular the critic learning rate tends to be reduced over time. If a certain hyperparameter was not important to agent performance we would expect less consistency in its evolution across seeds, as selection would be driven by other hyperparameters: thus indicating performance is more sensitive to critic learning rate than actor learning rate.

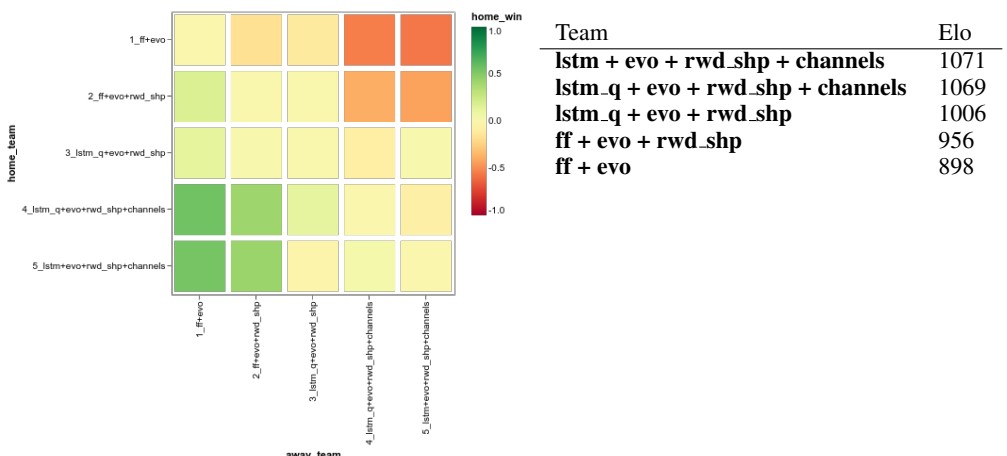

| Team | Elo |
|---|---|
| **lstm + evo + rwd_shp + channels** | 1071 |
| **lstm_q + evo + rwd_shp + channels** | 1069 |
| **lstm_q + evo + rwd_shp** | 1006 |
| **ff + evo + rwd_shp** | 956 |
| **ff + evo** | 898 |

Figure 10: Win rate matrix for the Tournament between teams: from top to bottom, ordered by Elo, ascending: **ff + evo**; **ff + evo + rwd_shp**; **lstm_q + evo + rwd_shp**; **lstm_q + evo + rwd_shp + channels**; **lstm + evo + rwd_shp + channels**. ELo derived from the tournament is given in the table.

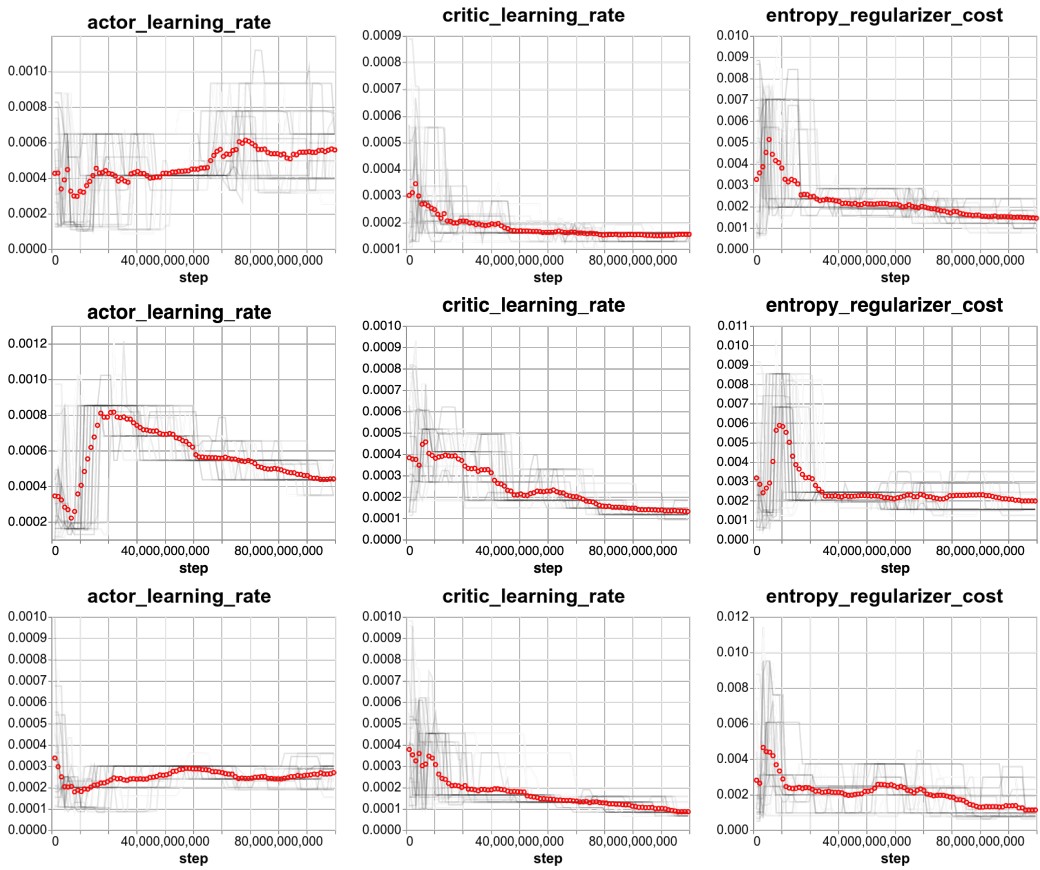

Figure 11: Hyperparameter evolution for three separate seeds, displayed over three separate rows.

## F  BEHAVIOR VISUALIZATIONS

As well as the videos at the website[3], we provide visualizations of traces of the agent behavior, in the repeated "cross pass" motif, see Figure 12.

---

[3]https://goo.gl/AuHv7V

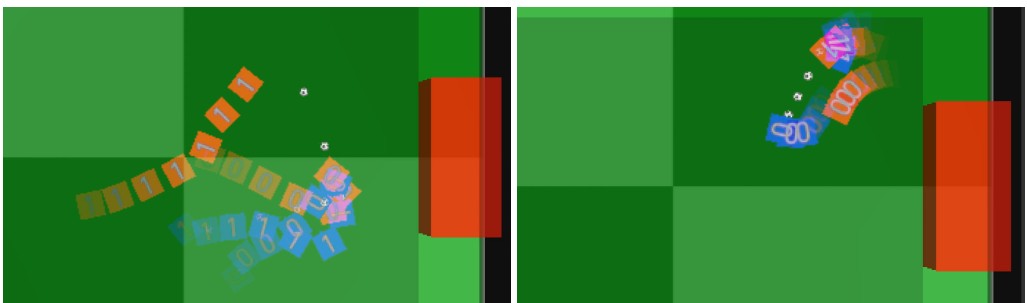

Figure 12: On the left red agent 0 has passed to agent 1, who apparently ran into position to receive. On the right blue agent 1 has passed to agent 0.

