# OpenReview forum: "Emergent Coordination Through Competition"
_ICLR.cc/2019/Conference_

### Official Review · AnonReviewer3 · 2018-11-03
**The paper presents a new simplified RoboCup environment that may be of some interest**

**Rating:** 6
**Confidence:** 3

**Review:**

This paper introduces a new multiagent research environment---a simplified version of 2x2 RoboSoccer using the MuJoCo physics engine with spherical players that can rotate laterally, move forwards / backwards, and jump.

The paper deploys a fine-tuned version of population-based sampling on top of a stochastic value gradient reinforcement learning algorithm to train the agents.  Some of the fine-tunings used include deploying different discount factors on multiple different reward channels for reward shaping.

The claimed novel contributions of the paper are (1) a new multiagent testbed, (2) a decentralized training procedure, (3) fine-tuning reward shaping, and (4) highlighting the challenges in evaluation in novel multiagent competitive environments.

Overall, my judgment is that the paper is fine, but the authors have not helped me to understand the significance of their contributions.

Taking each in turn:

(1) What is the significance of the new environment?  What unique characteristics make it difficult?  What makes this environment an importantly different testbed or development environment?  The connection to RoboSoccer is motivating but tenuous. The new environment should have particular characteristics that expose problems with past algorithms or offer new challenges existing algorithms have not addressed at all.

(2) Why is it important to have a decentralized training procedure when the authors have control over all the agents?  If it will allow faster training, has the authors' algorithm been demonstrated to accomplish that goal?

(3) It's hard to evaluate new algorithms when the domain studied is also new. We have no sense for state-of-the-art performance on this domain across a range of algorithms.  The authors conduct a careful ablation study on their new algorithm but do not compare their approach to other classes of algorithms.

(4) The authors indicate that evaluating the quality of an algorithm for a competitive context is hard in absence of established benchmarks---whereas in single-agent or cooperative environments progress can be measured against the goal of the environment, progress in competitive environments requires comparison to approaches that are thought to be good.  Here the authors are themselves pointing out a fundamental problem with introducing new competitive multiagent testbeds, and the authors don't resolve this tension.  Since the main contribution of the work is the environment, it's hard to see how this point the authors themselves make doesn't undermine that central contribution.

Besides other comments mentioned above, a couple other ways to improve the paper would be:
- Clarify why this environment is important to be introducing---what are the unique things that can be studied with this new environment?
- Hold an open competition to get benchmarks created by other teams of researchers

Some minor comments:
- $n_r$ is not defined explicitly in the text as far as I have found
- The authors state: "The specific shaping rewards use for soccer are detailed in Section 4.2" but I couldn't find them there.

---

Post-rebuttal

My main concern was assessing the value of the overall contribution of the paper. The other reviewers seem to appreciate both the new environment being offered and the combination of techniques deployed in the authors' solution. If there is an audience that will appreciate this work at ICLR as seems to be indicated by those reviews, then I would increase my score to marginally above the acceptance threshold.

---

> ### Author Response · Authors · 2018-11-16
> **author response to review 3**
>
> We thank the reviewer for constructive feedback. The contribution of our work extends beyond the introduction of a novel environment. We use the domain to study the emergence of coordination by analyzing the behaviors of decentralized agents. We carried out ablation studies to surface important ingredients for effective learning in multi-agent cooperative-competitive games. Our work highlights a fundamental difficulty in evaluation on multi-agent domains, with or without benchmarks, which we alleviate through a principled Nash averaging evaluation scheme.
>
> We address each point individually:
>
> 1) Q) “What makes this environment an importantly different testbed or development environment?”
> A) The environment will provide the ML community with a cooperative-competitive multi-agent environment in a simulated physical world which is accessible and flexible. It is accessible because it uses a widely adopted physics simulator and research platform. It is also accessible in the sense that we have demonstrated a solution using end-to-end RL. It is flexible because although the current paper describes a relatively simple agent embodiment (chosen to draw attention to multi-agent coordination), the environment can be extended in terms of body complexity as well as the number of players and could become part of a wider multi-task suite with consistent physics. We believe it is an important contribution to create such an environment, release it, and publish the first set of results on it. Further, the environment rules are simple but complexity emerges from sophisticated behavior and interactions between independent physically embodied agents. As such we have seen a level of emergent cooperation in a simulated physical world, which has not been witnessed before by end-to-end RL.
>
> Q) “The new environment should [...] offer new challenges existing algorithms have not addressed at all.''
> A) Learned cooperation of embodied independent RL agents in physical worlds is an unsolved problem, and a significant challenge for all existing approaches. To our knowledge there is no published environment that allows us to study this problem with realistic simulated physics where agents must acquire and leverage physical motor skills in order to coordinate with others in an open-ended manner.
>
> 2) Q) “Why is it important to have a decentralized training procedure when the authors have control over all the agents?”
> A) We agree that the environment could be used to investigate centralized approaches which could yield faster learning in this particular problem (but may not in general scale to more agents). However, we chose to study the emergence of coordination in decentralized, non-communicating agents, which is a significant unsolved problem important for real-world multi-agent problems (e.g. interaction between self-driving cars from different manufacturers, or human-agent interactions) where centralized solutions may not be feasible, and is more consistent with human learning.
>
> 3) Q) “It's hard to evaluate new algorithms when the domain studied is also new.” & “We have no sense for state-of-the-art performance on this domain across a range of algorithms”
> A) We agree that evaluation is difficult in the absence of clear baselines on a novel domain. We have combined state-of-the-art distributed RL and continuous control, with additional improvements, and suggest that this is a sensible reference solution for future investigations. We performed a detailed ablation study precisely to answer the question: what are the important ingredients for successful multi-agent learning on this novel, challenging domain?
>
> 4) Q) “The authors indicate that evaluating the quality of an algorithm for a competitive context is hard in the absence of established benchmarks”
> A) We disagree with reviewer’s assessment that highlighting difficulties in evaluation undermines the contribution of this work. There have been multiple studies (sec 4.3) where conclusions have been drawn according to simple multi-agent evaluation schemes. Our work shows where existing evaluation procedures fall short. We adopted an evaluation scheme via Nash averaging and demonstrated the discrepancy between our methods and a tournament (Figure 10). We do not claim that our evaluation method resolves the issue completely, but we believe it provides a more principled evaluation scheme. Even for domains where we possess human baselines or programmed bots evaluation is still difficult for the same underlying reason. It is important to introduce domains in which these problems arise, such as this one.
>
> Q) “what are the unique things that can be studied with this new environment?”
> A) See 1)
>
> Q) “Hold an open competition to get benchmarks created by other teams of researchers”
> A) we agree that our environment would be suitable for a competition, since the environment is an easily accessible MuJoCo environment. This could be an exciting future project, beyond the current paper scope.

---

### Official Review · AnonReviewer1 · 2018-11-05

**Rating:** 7
**Confidence:** 3

**Review:**

Summary: The authors use competition as a way to train agents in a complex continuous team-based control task: a 2 player soccer game. Agents are paired randomly into a team of 2 and play another team of 2. The key aspect of the proposed algorithm is the use of population based training.

Strong Points
-	The authors propose a convincing methodology for speeding up learning in coordinated MARL.
-	The Nash Averaging approach suggested for evaluating in the presence of cycles is interesting and a useful tool for evaluation when there are no easy baselines
-	The authors do convincing ablation studies to show that the PBT is the most important part of the learning algorithms and does well even when paired with a simple feed forward model

Questions
-	The authors use reward shaping of the form: “We design shaping reward functions {rj : S × A → R}j=1,...,nr P , weighted so that r(·) := nr j=1 αj rj (·) is the agent’s internal reward and, as in Jaderberg et al.” I’m not sure I follow how this works, without the additional dense shaping in the soccer game the reward is 0/1 depending on if one’s team wins or loses, so won’t one’s rewards always be perfectly correlated with those of one’s teammates and perfectly anticorrelated with those of the other team? Does this only work with the dense shaping (e.g. vel-to-ball)?
-	I would like to see which of the PBT controlled hyperparameters actually matter for the increase in training speed. Do the learning rates matter (since they’re also being changed by the Adam optimizer as training goes) or is it about the discount factor/entropy regularizer?

---

> ### Author Response · Authors · 2018-11-16
> **author response to review 1**
>
> We thank the reviewer for their constructive feedback. We address each point individually:
>
> Re. correlation of rewards within and across teams:
>
> In our setup we distinguish between the raw sparse reward events / raw continuous performance metrics (all denoted by r), and the individual agent’s preferences for these (denoted by alpha). While the binary reward events ‘goal’ and ‘concede’ are correlated within team, but anti-correlated across teams, this is not true for all continuous metrics (it is for ball-vel-to-goal but not for vel-to-ball). Independently, each agent can have different preferences for each of the signals and associated discount factors. These quantities are evolved via PBT and thus vary across agents and over time. As a consequence, even when the signal itself is perfectly (anti-)correlated between agents this is almost never true for the resulting reward received by the agents and they may thus acquire different behaviors.
>
> Re. relative importance of hyperparameter adjustments performed by evolution:
>
> The reviewer raised an important question regarding population-based training. Given that the PBT procedure drives evolution towards agents whose hyper-parameters and model parameters are the most competitive within the current population of agents (in terms of winning the game), a parameter that is irrelevant for the learning progress should not exhibit a consistent trend across experiment replicas (as each hyper-parameter is initialized randomly and then evolved through an evolution procedure that selects, inherits and mutates where mutation applies a random multiplicative perturbation). We concretely observed in our work (Figure 4) that both actor and critic learning rates as well as discount factor and entropy cost exhibit clear trends over the course of training. Regarding learning rates specifically, we believe that our PBT procedure re-discovers the commonly employed learning rate annealing schedule for accelerated learning. We have added a new Section E in the appendix comparing the evolution of hyperparameters across three experiments with different seeds: entropy cost and critic learning rates evolve consistently across experiments indicating that performance is more sensitive to these parameters. The critic learning rate in particular decreases over time. Actor learning rate is relatively less consistent across the three experiments, indicating that performance is less sensitive to fine tuning the actor learning rate.

---

### Official Review · AnonReviewer2 · 2018-11-07
**Well-written submission with good analysis**

**Rating:** 7
**Confidence:** 3

**Review:**

The paper proposes a new environment - 2vs2 soccer - to study emergence of multi-agent coordinated team behaviors. Learning relies on population-based training of agent's shaped reward mixtures and approach of nash averaging is used for evaluation.

Clarity: the paper is well-written and clear. The ablations provided are helpful in understanding how much different introduced components matter, and quantitative and qualitative analysis of resulting behavior is quite nice

Originality: the individual pieces of this work (PBT, SVG, nash averaging) have been introduced previously, but this paper puts them together in a well-chosen manner.

Significance: I believe this paper proposes a number of interesting observations (effects of PBT, evaluation, effects of recurrent policies to overcome non-stationarity issues) that I believe would be of value to the part of ICLR community doing research in multi-agent systems.

---

> ### Author Response · Authors · 2018-11-16
> **author response to review 2**
>
> We thank the reviewer for their constructive feedback.

---

### Meta-Review · Area_Chair1 · 2018-12-15
**An interesting new task to study learning cooperation between agents**

**Confidence:** 3
**Recommendation:** Accept (Poster)

**Metareview:**

The paper studies population-based training for MARL with co-play, in MuJoCo (continuous control) soccer. It shows that (long term) cooperative behaviors can emerge from simple rewards, shaped but not towards cooperation.

The paper is overall well written and includes a thorough study/ablation. The weaknesses are the lack of strong comparisons (or at least easy to grasp baselines) on a new task, and the lack of some of the experimental details (about reward shaping, about hyperparameters).

The reviewers reached an agreement. This paper is welcomed to be published at ICLR.